# Burnout and Psychological Vulnerability in First Responders: Monitoring Depersonalization and Phobic Anxiety during the COVID-19 Pandemic

**DOI:** 10.3390/ijerph19052794

**Published:** 2022-02-27

**Authors:** Veronica Benincasa, Maria Passannante, Filippo Perrini, Luna Carpinelli, Giuseppina Moccia, Tiziana Marinaci, Mario Capunzo, Concetta Pironti, Armando Genovese, Giulia Savarese, Francesco De Caro, Oriana Motta

**Affiliations:** Department of Medicine and Surgery, Baronissi Campus, University of Salerno, 84081 Baronissi, Italy; veronicabenincasa@alice.it (V.B.); mariapass@hotmail.it (M.P.); filippo.perrini@aslroma6.it (F.P.); lcarpinelli@unisa.it (L.C.); gmoccia@unisa.it (G.M.); tizianamarinaci@gmail.com (T.M.); mcapunzo@unisa.it (M.C.); cpironti@unisa.it (C.P.); armandogeno@gmail.com (A.G.); fdecaro@unisa.it (F.D.C.); omotta@unisa.it (O.M.)

**Keywords:** COVID-19 pandemic, first responders, helping professions, depersonalization, burnout, psychological vulnerability

## Abstract

Background: It is common knowledge that first responders are among the helping professionals most at risk of burnout and psychological vulnerability. During the COVID-19 pandemic, their mental health has been subjected to various risk factors. Methods: Data on socio-demographic characteristics, the Maslach Burnout Inventory (MBI) and psychological vulnerability (SCL-90-R) were obtained from 228 subjects (55.3% female; M age = 45.23, SD = 13.14) grouped on the basis of their actual involvement during the emergency phases (82% First Responders and 18% Second Responders). Results: First responders exceeded the MBI clinical cut-off, while SRs did not (χ² ≥ 0.5); specifically, EE = 89.8%, DP = 85.8%, and PA = 82.1%. The FR group showed a higher mean in the global severity index (GSI = 49.37) than did the SRs (=43.95), and the FR group exceeded the clinical cut-off in the SCL-90-R scales of SOM (51.06), ANX (52.40), and PHOB (53.60), while the SF group did so only for the PHOB scale (50.41). The MBI dimensions correlated significantly (*p* = 0.05) with all investigated clinical scales of the SCL-90-R. Conclusions: Emergency situations expose first responders to specific risk factors related to work performance and relational aspects, which contribute to increased psychological vulnerability and burnout.

## 1. Introduction

Several studies [1,2,3] have highlighted the strong impact of the COVID-19 pandemic situation on frontline workers, including first responders, identified as one of the highest-risk groups in terms of negative physical and mental health impacts since the beginning of the pandemic.

The definition of first responders typically includes professionals and/or volunteers specifically trained in traditional emergency response groups that include medical and paramedical rescue personnel, among others [4,5]. Specifically, these different groups, within their professional spheres, play different roles in response to a critical and emergency event and, consequently, are assisted by second responders who deal with the immediate restoration of procedures and protocols in the field, clinics, and hospitals.

Both first and second responder groups have in common the fact that they are among the first to participate in an emergency and often the very first to assist victims following a traumatic event [6], especially first responders; they are therefore usually exposed to emotionally demanding and unpredictable situations [7]. Within the health emergency, the growing need for social distance, the high risk of infection, and the pressures in interactions with an audience have been added, further exacerbating various aspects of the working life of professionals, especially in terms of emotional load. This is often translated into severe forms of burnout.

Burnout is generally defined as a syndrome of emotional exhaustion, depersonalization, and personal derealization, which can manifest itself in all those professions with very pronounced relational implications, primarily in helping professions [8]. 

## 2. Scientific Background 

The scientific interest in the burnout syndrome affecting health and parasanitary personnel has always been very broad and has broadened especially during the COVID-19 pandemic. Recent studies [1,2,3] have identified high burnout values and a prevalence of between 30% and 60% for frontline staff who faced the first stages of the pandemic. 

Baskin and colleagues [9] conducted a review of the literature, finding that around half of healthcare workers surveyed during the first wave of the COVID-19 pandemic reported moderate to high burnout, and healthcare workers who had higher burnout scores had reported lower resilience scores. Specifically, they highlighted statistically significant negative correlations between resilience and the “Emotional Exhaustion” (EE) and “Depersonalization” (DP) burnout subscales and a positive correlation between resilience and the “Personal Accomplishment” (PA) subscale. In addition, spending more than 50% of working time in contact with COVID-19 patients was associated with higher burnout scores among healthcare professionals than spending less than 25% of working time with COVID-19 patients.

There is the DP subscale, which can be hypothesized as an important variable to indicate major personal and environmental risk factors that could be remodeled as a resource for personnel. In fact, DP is characterized by both the work and personal factors of the subject [10]. Leiter and Harvie [11] found that DP can be understood as a way to address the exhaustion through which a worker attempts to gain emotional distance from the recipients of the service. Walkey and Green [12] found that DP always combined in a one-to-one way with EE to form the core of burnout.

In a study by Hu and colleagues [13], participants showed a moderate level of DP (42.3%) associated with high levels of fear (91.2%), anxiety (14.3%), and depression (10.7%). DP was negatively correlated with resilience, intra-family social support, and extra-family social support. Another study [14] identified chronic levels of burnout and consequent onset of state anxiety, acute stress, and symptoms of depersonalization/derealization in healthcare professionals engaged in the front line with patients with COVID-19. The dimension of psychological vulnerability in the operators themselves was also studied. Many studies [15,16,17,18,19,20,21], through the use of Symptom Checklist-90 revised (SCL-90-R) [22,23], have identified the parallel persistence of phobic anxiety (PHO) and psychoticism (PSY) symptoms among healthcare professionals during the COVID-19 pandemic.

In a study by Akan and colleagues [24], it was found that the PHO scale shows high clinical scores, along with depersonalization, in healthcare workers in the wards with the greatest risk of contracting the virus from COVID-19 patients.

As expressed in the manual of the SCL-90-R [23], the PHO clinical scale, which constitutes 7 of the 90 items of the test, is defined as a persistent fear response to a specific person, place, object, or situation that is characterized as irrational and disproportionate with respect to the stimulus. It leads to avoidance or flight behaviors. The PSY clinical scale includes items that evaluate a lifestyle that involves withdrawal and isolation, which could be the result of PHO linked to the fear of contacting the virus.

Psychoticism is one of the three traits used by the psychologist Hans Eysenck [25] in his P-E-N (Psychoticism, Extroversion, Neuroticism) personality model. Psychoticism refers to a personality pattern typical of aggression and hostility. The diagnostic psychiatric manual “DSM 5” [26] includes psychoticism in personality cluster B, characterized by characteristics of negative affectivity (experiencing intensely and frequently negative emotions) and detachment (withdrawal from other people and social interactions).

## 3. Purpose and Aims of the Study

Among the professional categories of first responders, health workers, in particular, are identifiable as those at greatest risk of exposure to the virus, and their commitment at the forefront of health emergency management involves increasing operational and emotional overload. 

It is evident, with reference to these premises confirmed by the scientific literature, that there is a physical dimension greater than the three dimensions of burnout in healthcare workers. However, we hypothesize that the cluster of first responders is further at risk, as they are involved not only in the routine of patient care, but under additional pressure from the confluent variables of urgency and emergency decision-making during critical events, as in the current case of the COVID-19 pandemic.

The objective of this study, therefore, was to: (a) Assess the burnout levels in health professionals involved in the frontline care of COVID-19 patients; (b) monitor the levels of depersonalization, phobic anxiety, and psychoticism in relation to trends of increase and/or decrease in hospitalizations of COVID-19 patients; and (c) verify any correlations due to personal or work context variables intended as risk factors and maintenance of the clinical symptoms detected.

## 4. Materials and Methods

### 4.1. Participants

The health personnel involved during the COVID-19 pandemic was 300 units.

In total, 228 subjects participated (M age = 45.23; SD = 13.14), of which 55.3% were female. 

The socio-demographic characteristics (see Table 1) were as follows: 59.6% of the sample were married, 80.7% had a permanent contract, and 59.6% had worked for more than 10 years at the hospital. 

In order to support the increase in COVID-19 hospitalizations, at the end of April 2020 a “Covid Center” was set up at the “A.O.U. San Giovanni di Dio e Ruggi d’Aragona”, in which n° 24 hospital beds were added to the n° 16 already present in the Intensive and Sub-Intensive Care Unit. 

The hospital units represented in the full sample were Infectious Diseases (14.9% of participants), Emergency Medicine (10.1%), Emergency Room (23.2%), Anesthesiology and Surgery Care (23.7%), Intensive Care (10.1%), and Cardiology (18%). 

In addition, the medical and paramedical operators of the hospital units were further grouped on the basis of their actual involvement during the emergency phases into 82% first responders (FRs) and 18% second responders (SRs).

### 4.2. Procedures

This study involved the health and parasanitary personnel of the “A.O.U. San Giovanni di Dio and Ruggi D’Aragona” (Salerno, Italy) in two specific moments connected to an increase and a decrease in hospitalizations due to the COVID-19 pandemic. 

So, a first evaluation (T0) was conducted during the so-called “first wave” phase of the pandemic (the period of February–May 2020) in which a high number of hospitalizations was recorded (specifically, n°124 COVID-19 patients were hospitalized); while further monitoring (T1) was carried out in the months June–September 2020 (this period so-called “Second wave“), in which there was a decrease in hospitalizations (n°10 of COVID-19 patients were recorded) and a reduction of ministerial restrictions.

### 4.3. Data Collection Tools

The survey was conducted through the use of two standardized and validated tests used in the clinical and research fields. Specifically, the following were administered:

1. Symptom Checklist-90-R (SCL-90-R) [22,23], a self-report questionnaire consisting of 90 items that assess 9 primary symptom dimensions: Somatization (SOM), Obsessive–Compulsive (O-C), Interpersonal Sensitivity (IS), Depression (DEP), Anxiety (ANX), Hostility (HOS), Phobic Anxiety (PHOB), Paranoid Ideation (PAR), and Psychoticism (PSY). It also investigates 7 additional items (OTHER) that evaluate appetite and sleep disorders and a further 3 global indices: the Global Severity Index (GSI): a global indicator of the current intensity of mental distress perceived by the subject; the Positive Symptom Total (PST): reflects the number of symptoms reported by the subject; and the Positive Symptom Distress Index (PSDI): a response style index.

The severity of each item is rated on a 5-point Likert scale ranging from “1 = not at all” to “5 = very much”. The total score is the sum of the scores of the 90 items, and the average score of each factor equals the total score of the items included in the factor subscale divided by the number of items. The standardized clinical cut-off of 50% was used to assess the presence of psychological vulnerability.

Considering the purposes of this work and the correlation indices already present in the literature, the SOM, DEP, ANX, HOS, PHOB, and PSY scales and the GSI index were taken into consideration. 

The degree of internal consistency, calculated on the total scores of the subscales of the SCL-90-R, indicates a coefficient α of 0.93, showing excellent reliability of the test.

2. Maslach Burnout Inventory (MBI) [27,28], a questionnaire of 22 items, each with 6 degrees of response on a 6-point Likert scale, designed to assess the level of burnout of an individual. The MBI is a multidimensional questionnaire that addresses three different fields of professionalism: (1) Emotional Exhaustion (EE): examines the feeling of being emotionally parched and exhausted from one’s work; (2) Depersonalization (DP): measures a cold and impersonal response towards the users of one’s service, care, treatment, or performance; and (3) Personal Accomplishment (PA): evaluates the feeling of one’s competence and one’s desire for success in working with others.

The questionnaire offers a quantitative assessment by identifying three degrees of severity: high, medium, and low.

The reliability calculated via the “retest” method was satisfactory, with coefficients ranging between 0.70 and 0.87.

### 4.4. Data Analysis

Data were analyzed using the “IBM SPSS Statistics 23.0 software package“ (SPSS^®^ Statistics, Chicago, IL, USA). Data conforming to the normal distribution in descriptive statistics are presented as the mean (M) ± standard deviation (SD). A comparison was made between the means of the standardized scores obtained in the SCL-90-R and MBI tests. Cross-tabs were carried out based on the most significant variables, namely, the affiliation to hospital departments and length of service.

Differences in the numerical data between two sample groups were analyzed via ANOVA test.

## 5. Results

### 5.1. Burnout Levels

As regards the burnout levels present in the groups, Table 2 shows that in phase T0, the FRs on average exceeded the clinical cut-off, unlike the SRs (χ² ≥ 0.5), in the three dimensions of the MBI; specifically, EE = 89.8%, DP = 85.8%, and PA = 82.1%. 

Furthermore, there was a difference between the means of the scores obtained by the two groups. Specifically, at T0, the average score for the FR group in EE was 14.40 (SRs = 9.68; F = 0.006), that in DP was 8.50 (SRs = 5.76; F = 0.005), and that in the PA dimension was 25.35 (SRs = 19.78; F = 0.002).

In phase T1, the FR group maintained a score of >50% in the three dimensions of burnout, unlike the SRs, while there was a decrease in the mean scores obtained on the MBI: for EE, 13.28 (SRs = 10.61; F = 0.112); for DP, 7.61 (SRs = 5.37; F = 0.029); and for the PA dimension, 24.03 (SRs = 19.76; F = 0.037).

### 5.2. Psychological Vulnerability

To assess psychological vulnerability, the means of the standardized SCL-90-R clinical scales scores at both T0 and T1 were compared (see Table 3). From the analysis of the results obtained at T0, it emerged that neither group exceeded the clinical cut-off of the GSI scale, which evaluates the global symptom index, though the FR group showed a higher mean (GSI = 49.37) than the SRs (GSI = 43.95). Furthermore, the FR group exceeded the clinical cut-off in the SOM (51.06), ANX (52.40), and PHOB (53.60) scales, while the SR group did so only for the PHOB scale (50.41).

In phase T1, the means of the scores underwent a decrease in both groups investigated, and the clinical scales that in phase T0 exceeded the clinical cut-off were within the standardized mean, although the result is that the FRs showed higher vulnerability levels compared to SRs. 

### 5.3. Correlations

From the bivariate correlational analysis (*p* = 0.05) carried out between the context and personal variables of the sample and the indices of psychological vulnerability and the dimensions of burnout, it emerged that gender correlated with the SCL-90-R scales of SOM (*p* = 0.160), PSY (*p* = 0.144), and GSI (*p* = 0.131) and with PA of the MBI (*p* = 0.191). Length of service correlated with gender (*p* = 0.147), age (*p* = 0.569), and the HOS scale (*p* = 0.151). The type of hospital unit correlated with the FR category (*p* = 0.700) and with PA (*p* = 0.154). The Responders category showed significant positive and negative correlations (*p* = 0.05) with the MBI dimensions (EE = −0.183; DP = 0.186; PA = 0.201) and with the clinical scales of the SCL-90-R (*p* = 0.01) investigated (SOM = −0.160; DEP = −0.177; ANX = −0.192; HOS = −0.149; PSY = −0.139; GSI = −0.190).

The MBI dimensions correlated significantly (*p*= 0.05) with all investigated clinical scales of the SCL-90-R.

## 6. Discussions

This work, through the results collected, supports some of the data already present in the literature; specifically, it can be a source of reflection with respect to intervention programs and protocols useful for restoring the personal resources of health and parasanitary workers. 

As can be seen from the results of the MBI, the FR (first responder) group was more exposed to burnout than the SR (second responder) group, as noted in the study by Baskin and colleagues [29].

In particular, interesting data were obtained regarding the DP dimension. FRs showed higher mean scores than SRs at T0, and SRs showed an increase in mean scores at T1 [13]. By the term "depersonalization", Maslach [28] means an attitude characterized by detachment and hostility that primarily involves the professional helping relationship, experienced with annoyance, coldness, and cynicism. Consequently, the subject tries to avoid involvement, limiting the quantity and quality of their professional interventions to the point of evasively responding to requests for help and underestimating or denying the patient’s problems. Recent studies [13,14] have shown a correlation between high levels of DP and anxious–depressive symptoms; in fact, a significant correlation emerged from the analysis of our data (*p* = 0.05) between the DP dimension and the clinical scales of ANX (*p* = 0.384) and DEP (*p* = 0.375), as well as the scales SOM (*p* = 0.364), HOS (*p* = 0.412), PHOB (*p* = 0.256), and GSI (*p* = 0.413). These correlations find support in the definition of burnout, which presents various symptoms that can be classified into: non-specific (restlessness, sense of tiredness, exhaustion, apathy, nervousness, and insomnia), somatic (ulcers, headaches, weight gain or loss, nausea, cardiovascular disorders, and sexual difficulties), and psychological (depression, low self-esteem, guilt, feelings of failure, anger, resentment, irritability, aggression, high resistance to going to work, indifference, negativism, isolation, suspicion and paranoia, rigidity of thinking and resistance to change, difficulties in relationships with users, cynicism, and guilty attitude towards users and work colleagues) [30]. From a clinical point of view, the symptoms of burnout are many; they recall the anxiety–depressive spectrum disorders and underline the particular tendency to somatization and the development of behavioral disorders, and the symptomatological correlation with conditions of distress is, in any case, strong. In support of this, significant correlations also emerged between the EE and PA dimensions of the MBI and the aforementioned clinical scales of the SCL-90-R.

In consideration of the socio-demographic and contextual variables of the operators, it is important to highlight how the length of service correlates with the HOS scale of the SCL90-R (*p* = 0.151), while belonging to the relevant department correlates with the magnitude of PA on the MBI (*p* = 0.154). Professional performance is conditioned by the presence of numerous risk factors, whether personal, such as the demographic variables, personality characteristics, expectations, and values of the individual, or organizational, such as overload of work, lack of personnel and/or demotion, working and organizational conditions, and a low level of both economic satisfaction and recognition of skills. The continuous presence of these variables can contribute to increasing levels of both mental and physical stress in the professional field [31,32], with relapses of a psychosomatic nature, including emotional and behavioral fragility; these relapses during professional activity [33,34,35] may contribute to an increased risk of mistakes [36] and may consequently affect the patient-perceived quality of care [31,32,36].

## 7. Conclusions

In general, emergencies expose healthcare and paramedical personnel to a series of specific risk factors related to the care of the infected patient, but also to substantial changes in the work as regards organizational, relational, and safety-related aspects, which contribute to an increase in psycho-physical stress and burnout. 

Specifically, the prolongation of the health emergency due to COVID-19 has led to an increase in pressure and fear and, in some cases, has led to a chronicization of psychological vulnerabilities; if prolonged over time and accompanied by high intensity, this can determine exhaustion of personal resources, in some cases favoring the appearance or chronicization of burnout. The monitoring carried out in conjunction with the increase/decrease in hospitalizations due to COVID-19 highlighted an already precarious symptomatic situation present above all in FRs, who, in urgent and emergency situations, are called to solve the problems and inconveniences of their patients, trying to implement coping strategies useful to the situation.

For these reasons, the topicality of the issue of health protection for health professionals in relation to the COVID-19 emergency is clear and relevant, more specifically with regard to mental health. 

It is therefore important to implement specific policies, especially preventive ones. In addition to alleviating anxiety and depression, which are necessary to improve EE and DP levels, work stress is one of the factors that influences personal results. It is essential to establish a policy that reflects these variables. To this end, health facilities must provide measures capable of analyzing and regulating these conditions. It would be useful to introduce educational programs to improve and strengthen ways to proactively counter and regulate stress, resulting from workload and interpersonal relationships, during the pandemic (but it suffers for the lack of data proceeding the COVID-19 pandemic) [37,38,39,40,41].

This study has the following limitations. First, the investigation is limited in monitoring a single hospital that adopted specific protocols during the COVID-19 pandemic that are different to other national hospitals. Second, this study ruled out some factors that can affect hospital employee burnout during a pandemic such as COVID-19. For example, the lack of prior data in the COVID-19 period. Furthermore, numerous psychological, physical, and environmental factors were not included in the data collection. This could be a limit in the qualitative and quantitative evaluation to be taken into account.

## Figures and Tables

**Table 1 ijerph-19-02794-t001:** Socio-demographic and work context characteristics of the total sample.

Main Categories	Variable		Percent (%)
Socio-demographic	Gender	Female Male	55.3 44.7
	Marital status	Single Married Separated/Divorced Widower	30.7 59.6 7.4 0.4
Employment	Seniority	<10 years >10 years	40.4 59.6
	Contract type	Permanent Fixed-term	80.7 19.3
	Hospital unit	Infectious Diseases Emergency Medicine Emergency Room Anesthesiology/Surgery Care Intensive Care Cardiology	14.9 10.1 23.2 23.7 10.1 18
	Group	First Responders Second Responders	82 18

**Table 2 ijerph-19-02794-t002:** Differences between groups in terms of mean scores in the three dimensions of the MBI in phase T0.

		Emotional Exhaustion (EE)		Depersonalization (DP)		Personal Accomplishment (PA)	
		T0	T1	T0	T1	T0	T1
First Responders (FR)	Mean SD	14.40 10.22	13.28 9.62	8.50 5.80	7.61 6.07	25.35 9.71	24.03 11.40
Second Responders (SR)	Mean SD	9.68 7.51	10.61 10.09	5.76 4.59	5.37 5.05	19.78 13.40	19.76 13.67

**Table 3 ijerph-19-02794-t003:** Differences between groups in terms of mean scores on SCL-90-R scales.

		SOM		DEP		ANX		HOS		PHOB		PSY		GSI	
		T0	T1	T0	T1	T0	T1	T0	T1	T0	T1	T0	T1	T0	T1
FRs	Mean SD	51.06 11.52	46.17 8.89	49.68 10.52	45.22 8.34	52.40 11.20	46.58 8.97	48.45 9.57	44.53 6.97	53.60 10.58	49.66 8.70	49.23 8.99	46.66 7.38	49.37 11.30	44.17 9.14
SRs	Mean SD	46.34 9.74	43.22 8.68	45.05 6.71	42.27 5.56	46.98 7.79	44.05 8.97	44.76 7.89	43 7.20	50.41 9.31	47.22 6.53	46.15 5.7	44.37 5.35	43.95 8.18	40.93 6.90

## Data Availability

Written informed consent was obtained from the subject(s) in order to publish this paper.

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
