# Peer review of "Burnout and Psychological Vulnerability in First Responders: Monitoring Depersonalization and Phobic Anxiety during the COVID-19 Pandemic"

_ijerph, 2022, doi:10.3390/ijerph19052794_

Round 1

Reviewer 1 Report

Dear Authors,

Thank you for submitting this manuscript, which addresses, in an original manner, a very interesting and timely topic. I enclose some suggestions for improving your paper.

  • I would include the paragraph “Purpose and aims of the study” in the Introduction;
  • I would add a specific paragraph dedicated to a literature review, including some of the contents already presented in the introduction and specifying why your paper is original with respect to what has been already written on the topic;
  • I would include the paragraph “Participants” in the Methodology;
  • In conclusions, please specify the practical/policy implications of your work; moreover, I would address the issue of the generalizability of results, also in light of the limitations of the study, to be discussed too.

Best wishes!

Author Response

Rew 1

Comments and Suggestions for Authors

Dear Authors,

Thank you for submitting this manuscript, which addresses, in an original manner, a very interesting and timely topic. I enclose some suggestions for improving your paper.

  • I would include the paragraph “Purpose and aims of the study” in the Introduction;
  • I would add a specific paragraph dedicated to a literature review, including some of the contents already presented in the introduction and specifying why your paper is original with respect to what has been already written on the topic;
  • I would include the paragraph “Participants” in the Methodology;
  • In conclusions, please specify the practical/policy implications of your work; moreover, I would address the issue of the generalizability of results, also in light of the limitations of the study, to be discussed too.

Best wishes!

Authors: The Authors would like to thank for the feedback and suggestions received which have certainly improved our paper. Changes are highlighted in yellow.

-  We have included subsection 1.1 "Purpose and aims of the study" in the "Introduction" paragraph and clarified our purpose.

- We have remodeled some studies in the scientific literature and expanded the impact of our research scope.

- We have included the paragraph “Participants”

- In the "Conclusions" paragraph we have included the implications of our study and we have also included a "Limitations" paragraph.

Reviewer 2 Report

This article has great merit in terms of what it offers to the field of public health and specifically the study of mental health for first responders during the COVID-19 pandemic. A few specific comments by section:

  • The introduction provides a solid overview of existing literature, though I would encourage you to tighten the writing in the first paragraph (it is a little wordy), to clarify the last sentence in paragraph 3 (the use of "more or less" is confusing), and to clarify the use of EE and DP in existing studies (the 6th paragraph claims most studies focus on EE, yet the examples suggest otherwise).
  • The methods were clearly discussed, well executed, and visualized effectively in the tables.
  • The discussion was also clear in explicating the data, though I would encourage you to consider adding a brief comment on the limitations of your study.
  • Finally, the conclusion was clearly written; here I would recommend adding a brief commentary on implications, which the final paragraph starts to move toward but does not explicitly articulate. 

I want to emphasize that the study design and article are sound, and these recommendations are meant to enhance the good work you've already done.

Author Response

Comments and Suggestions for Authors

This article has great merit in terms of what it offers to the field of public health and specifically the study of mental health for first responders during the COVID-19 pandemic. A few specific comments by section:

  • The introduction provides a solid overview of existing literature, though I would encourage you to tighten the writing in the first paragraph (it is a little wordy), to clarify the last sentence in paragraph 3 (the use of "more or less" is confusing), and to clarify the use of EE and DP in existing studies (the 6th paragraph claims most studies focus on EE, yet the examples suggest otherwise).
  • The methods were clearly discussed, well executed, and visualized effectively in the tables.
  • The discussion was also clear in explicating the data, though I would encourage you to consider adding a brief comment on the limitations of your study.
  • Finally, the conclusion was clearly written; here I would recommend adding a brief commentary on implications, which the final paragraph starts to move toward but does not explicitly articulate.

I want to emphasize that the study design and article are sound, and these recommendations are meant to enhance the good work you've already done.

Authors: The Authors thank for the positive feedback received and for the detailed reading carried out by the Reviewer who has grasped the importance of our work. For the valuable suggestions, all changes are underlining in green as follow:

- In the paragraph "Introduction" we have eliminated some repetitive sentences in order to make the text more discursive; we have clarified the sentence of paragraph 3 and reformulated the discourse of the scientific literature on DP and EE.

- We have clarified the implications of our findings in the "Conclusions" section.

- We have inserted the "Limitations" paragraph.

Round 2

Reviewer 1 Report

Dear Authors,

Thank you for revision the paper, which has consistently improved with respect to the previous version. Before accepting the manuscript, however, I suggest you the following modifications to further improve your work:

  • I would not separate both introduction and conclusions into sub-sections/sub-paragraphs.
  • A literature review section is still missing. If possible, try to include it in a separate paragraph.
  • Please correct typos (e.g.: "partecipants").

Best wishes!

Author Response

  The Authors thank the Reviewer for the effort and time dedicated to improving our paper with valuable suggestions. In the latest version of the paper we have standardized the "Introduction" paragraph, inserted the "Scientific background" paragraph and standardized the limitations and conclusions in a single "Conclusions" paragraph. We have corrected any spelling errors. The changes are in green. Thank you!